SciPost Physics

Submission

# Implications of new physics in $\Lambda_b \rightarrow \Lambda_c \ell\, \nu_\ell$ decay processes

Aishwarya Bhatta[*]

University of Hyderabad, Hyderabad, India-500046
*aish.bhatt@gmail.com

February 11, 2025

*16th International Workshop on Tau Lepton Physics (TAU2021),*
*September 27 – October 1, 2021*

## Abstract

Several indications of lepton non universality ratios, $R(D^*)$, $R(J/\Psi)$ and the measurements on hadronic and $\tau$ longitudinal polarizations in $b \rightarrow c\ell\, \bar{\nu}_\tau$ processes have attracted a lot of attentions. By considering the most general effective Lagrangian, we carry out a model independent analysis of the semileptonic $\Lambda_b$ decays, to inspect the nature of new physics. We constraint the new physics parameter space by using the chi square fitting. We study the implications of constrained new couplings on the observable such as branching fractions, forward-backward asymmetries, lepton non universality parameter and $\Lambda_c$ and lepton longitudinal polarization fractions of the decay modes.

## 1 Introduction

Fascinating hints of distinctness between the standard measured data and the Standard Model (SM) prognostications have been noticed in B decays by many experimental collaborations [1,2]. Those observations can be treated as indirect substantiation of physics beyond the SM which have drawn awareness by the scientific community in recent years. Amid these decays,

the charged-current modes are of special concern. Regardless of being a semileptonic $b \to c \tau \bar{\nu}$ channel, which are taking at tree-level in the SM, various experiments have announced substantial tensions in the branching fraction ratios ($\mathcal{B}$) with $\ell = e$ or $\mu$ [3–6]

$$R_{D^{(*)}} = \frac{\text{Br}(\bar{B} \to \bar{D}^{(*)} \tau \bar{\nu}_\tau)}{\text{Br}(\bar{B} \to \bar{D}^{(*)} l \bar{\nu}_l)} \tag{1}$$

$$R_{J/\psi} = \frac{\text{Br}(B_c \to J/\psi \tau \bar{\nu}_\tau)}{\text{Br}(B_c \to J/\psi l \bar{\nu}_l)} \tag{2}$$

These fractions are especially explorations of New Physics (NP) due to the abandonment of incertitude inherent in idiomatic $\mathcal{B}$ predictions. In contrast, at LHCb nearly 20% from the total number of produced hadrons are $\Lambda_b$ baryon, because of that the research of $\Lambda_b$ becomes quite enthralling presently. We explore the existence of lepton universality violation in tune with the baryon decays $\Lambda_b \to \Lambda_c l \bar{\nu}_l$ to confirm the effect and the presence of NP in the $B$ sector.

Table 1: Catalogue of measured lepton non-universality variables.

| LNU parameters | Experimental value | SM prediction | Deviation |
|---|---|---|---|
| $R_D$ | $0.391 \pm 0.041 \pm 0.028$ | $0.300 \pm 0.008$ | $1.9\sigma$ |
| $R_{D^*}$ | $0.316 \pm 0.016 \pm 0.010$ | $0.252 \pm 0.003$ | $3.3\sigma$ |
| $R_{J/\psi}$ | $0.71 \pm 0.17 \pm 0.184$ | $0.289 \pm 0.01$ | $2\sigma$ |

## 2 Theoretical framework and Observables

### 2.1 Effective Lagrangian

The constructive Lagrangian related with $\Lambda_b \to \Lambda_c l \bar{\nu}_l$ decay processes, refered by the quark transition $b \to c l \bar{\nu}_l$ is [7]

$$\mathcal{L}_{\text{eff}} = -\frac{4 G_F}{\sqrt{2}} V_{cb} \left\{ V_R \bar{l}_L \gamma_\mu \nu_L \bar{c}_R \gamma^\mu b_R + (1 + V_L) \bar{l}_L \gamma_\mu \nu_L \bar{c}_L \gamma^\mu b_L \right.$$

$$\left. + S_R \bar{l}_R \nu_L \bar{c}_L b_R + S_L \bar{l}_R \nu_L \bar{c}_R b_L \right\} + \text{h.c.} \,, \tag{3}$$

In the existence of NP, the differential decay distribution for $\Lambda_b \to \Lambda_c l \bar{\nu}_l$ process with respect to $\cos\theta$ and $q^2$ is given as [8]

$$\frac{d\Gamma}{dq^2} = = \frac{G_F^2 |V_{qb}|^2 q^2 \sqrt{\lambda}}{2^{10} \pi^3 M_{B_1}^3} \left( 1 - \frac{m_l^2}{q^2} \right)^2$$

$$\left[ A_1 + \frac{m_l^2}{q^2} A_2 + 2 A_3 + \frac{1}{4} A_4 + \frac{4 m_l}{\sqrt{q^2}} (A_5 + A_6) + A_7 \right], \tag{4}$$

The helicity amplitudes in terms of the various form factors and the NP couplings are given as [8]. After consolidating out $\cos\theta$ in Eqn. (4), we get the $q^2$ reliant differential decay rate. Some other engrossing parameters are shown below, apart from the branching ratios, in these decay modes.

## 2.2 Observables

- Forward-backward asymmetry parameter [10]:

$$A_{FB}(q^2) = \left( \int_{-1}^{0} \frac{d^2\Gamma}{dq^2 d\cos\theta_l} d\cos\theta_l - \int_{0}^{1} \frac{d^2\Gamma}{dq^2 d\cos\theta_l} d\cos\theta_l \right) \bigg/ \frac{d\Gamma}{dq^2} \,. \tag{5}$$

- Longitudinal hadron polarization asymmetry parameter [10]:

$$P_L^{\Lambda_c}(q^2) = \frac{d\Gamma^{\lambda_c=1/2}/dq^2 - d\Gamma^{\lambda_c=-1/2}/dq^2}{d\Gamma/dq^2} \,, \tag{6}$$

- Lepton non-universality parameter [10]:

$$R_{\Lambda_c} = \frac{\mathrm{Br}(\Lambda_b \to \Lambda_c \tau^- \bar{\nu}_\tau)}{\mathrm{Br}(\Lambda_b \to \Lambda_c \, l^- \bar{\nu}_l)}, \quad l = e, \mu. \tag{7}$$

## 3 Constraints on New Couplings

After assembling the expressions for all the interesting observables in presence of NP, we perform the $\chi^2$ fitting to obtain the values of the new coefficients, from the observables $R_{D^{(*)}}$, $R_{J/\psi}$ and $\mathrm{Br}(B_c^+ \to \tau^+ \nu_\tau)$ [9], where $\chi^2$ is defined as

$$\chi^2(X) = \sum_i \frac{(\mathcal{O}_i^{\mathrm{th}}(X) - \mathcal{O}_i^{\mathrm{Expt}})^2}{(\Delta\mathcal{O}_i^{\mathrm{Expt}})^2 + (\Delta\mathcal{O}_i^{\mathrm{SM}})^2} \,. \tag{8}$$

Here $\Delta\mathcal{O}_i^{\mathrm{SM}}$ and $\Delta\mathcal{O}_i^{\mathrm{Expt}}$ are respectively the SM and experimental uncertainties of the parameters. $\mathcal{O}_i^{\mathrm{th}}(X)$ are the total theoretical predictions for the observables with $X(= V_{L,R}, \ S_{L,R})$ as the new Wilson coefficients and $\mathcal{O}_i^{\mathrm{Expt}}$ represent the corresponding measured central values.

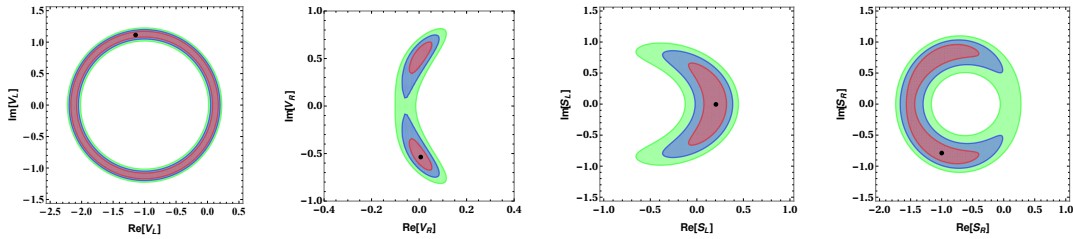

Figure 1: Constraints on various new coefficients obtained from $\chi^2$ fitting. Here red, blue and green colours represent the $1\sigma$, $2\sigma$ and $3\sigma$ contours and the black dots represent the best-fit values.

## 4 Results and Discussion

In the Table:3 we show the numerical results for $\Lambda_b$ decay with third generation leptons in the concluding state. We convey the differential branching ratio in Fig. 2, forward-backward asymmetry in Fig. 3, longitudinal hadron polarization in Fig. 4 and lepton non-universality in Fig. 5 for $\Lambda_b \to \Lambda_c \tau^- \bar{\nu}_\tau$ process with respect to $q^2$. In these figures, the blue dashed lines represent the SM contribution, the red, blue, green and yellow bands are due to the presence

Table 2: Predicted best-fit values of new Wilson coefficients with the $\chi^2_{\min}/\text{d.o.f}$ and pull values.

| New Wilson coefficients | Best-fit values | $\chi^2_{\min}/\text{d.o.f}$ | Pull |
|---|---|---|---|
| $(\text{Re}[V_L], \text{Im}[V_L])$ | $(-1.223, 1.035)$ | 1.151 | 2.982 |
| $(\text{Re}[V_R], \text{Im}[V_R])$ | $(-0.0044, -0.3283)$ | 1.145 | 2.984 |
| $(\text{Re}[S_L], \text{Im}[S_L])$ | $(0.965, 0)$ | 4.213 | 1.663 |
| $(\text{Re}[S_R], \text{Im}[S_R])$ | $(-0.694, -0.777)$ | 2.175 | 2.616 |

Table 3: The predicted values of branching ratios, forward backward asymmetry, longitudinal hadron polarization asymmetry and lepton non-universality parameters of $\Lambda_b \to \Lambda_c \tau \bar{\nu}_\tau$ processes in the SM and in the presence of $V_{L,R}$ and $S_{L,R}$ coefficients.

| Observables | SM prediction | Values for $V_L$, $V_R$ coupling | Values for $S_L$, $S_R$ coupling |
|---|---|---|---|
| $\text{Br}(\Lambda_b \to \Lambda_c^+ \tau^- \bar{\nu}_\tau)$ | $1.76 \times 10^{-2}$ | $4.83 \times 10^{-2}, \quad 2.25 \times 10^{-2}$ | $1.96 \times 10^{-2}, \quad 2.05 \times 10^{-2}$ |
| $A_{FB}$ | $-0.09$ | $-0.09, \quad -0.15$ | $-0.11, \quad -0.01$ |
| $P_L^{\Lambda_c}$ | $-0.795$ | $-0.795, \quad -0.441$ | $-0.782, \quad -0.437$ |
| $R_{\Lambda_c}$ | $0.352$ | $0.965, \quad 0.452$ | $0.393, \quad 0.411$ |

of $V_L, V_R, S_L$ and $S_R$ NP coefficients respectively. The branching ratios of $\Lambda_b \to \Lambda_c \tau^- \bar{\nu}_\tau$ deviate significantly from their corresponding SM values due to the NP contribution. For the forward-backward asymmetry and hadron polarization asymmetry, we found no deviation from SM results in the presence of $V_L$ but for $V_R$, $S_L$ and $S_R$ it give significant contributions to NP. For lepton non-universality parameter, We observe that all the coefficients show some deviation from SM, but the NP contribution coming from the $V_L$ coupling has significant impact on $R_{\Lambda_c}$ [10].

# 5   Conclusion

We have performed a model independent analysis of baryonic $\Lambda_b \to \Lambda_c \tau \bar{\nu}_\tau$ decay process by considering the generalized effective Lagrangian in the presence of new physics. Using the allowed parameter space, we estimated the branching ratios, forward-backward asymmetry, longitudinal hadron polarization and LNU parameter. To conclude, we have explored the effect of individual complex $V_{L,R}$ and $S_{L,R}$ couplings on the angular observables and found profound deviation from the standard model results due to the presence of these NP couplings.

# Acknowledgements

AB would like to thank R. Mohanta for helpful discussions. This work is supported by DST-INSPIRE(DST), Government of India, Ministry of Science and Technology for financial support through grant No. DST/INSPIRE Fellowship/2017/IF170644.

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
