# Peer review of "Delving into new physics in semileptonic $b \to c τ\bar ν$ transitions"

_SciPost Physics Proceedings_

## Round 1 · Referee Report · Anonymous (Referee 1) · 2022-4-11

Strengths

  • none

Weaknesses

  • if truly a proceeding, no reference to original work or any future work
  • incomplete description of the analysis (likelihood, observables)
  • incorrect treatment of hadronic uncertainties

Report

Given the lack of reference to an original work or future work, I must assume that this is an attempt to shoehorn an analysis into a conference's proceedings. The description of the analysis is incomplete, starting with the lack of chi2 and dof, both of which are needed to compute a p value. (Only their ratio is given). There is no statement on how the "SM uncertainty" is treated, which I assume is the hadronic uncertainty in the observables. The author implicitly uses a scheme for the treatment of theory uncertainties that regularly fails for LFU searches: the hadronic uncertainties are functions of the BSM parameter (here, the Wilson coefficients). This makes their treatment as part of the chi^2 ill-defined. They must be treated as nuisance parameters instead.

Requested changes

Assuming this will be resubmitted as a paper rather than a proceedings article, I recommend to - rework the treatment of the hadronic uncertainties as nuisance parameters with suitable theory constraints (e.g., gaussian priors from lattice QCD) - be clear which parameters are varied in which scenario

From that point forward, one can discuss suitability for a journal.

---

## Editorial Decision

accepted_in_target_journal